# MISE: Meta-knowledge Inheritance for Social Media-Based Stressor Estimation

## ABSTRACT

Stress haunts people in modern society, which may cause severe health issues if left unattended. With social media becoming an integral part of daily life, leveraging social media to detect stress has gained increasing attention. While the majority of the work focuses on classifying stress states and stress categories, this study introduce a new task aimed at estimating more specific stressors (like exam, writing paper, etc.) through users' posts on social media. Unfortunately, the diversity of stressors with many different classes but a few examples per class, combined with the consistent arising of new stressors over time, hinders the machine understanding of stressors. To this end, we cast the stressor estimation problem within a practical scenario few-shot learning setting, and propose a novel meta-learning based stressor estimation framework that is enhanced by a meta-knowledge inheritance mechanism. This model can not only learn generic stressor context through meta-learning, but also has a good generalization ability to estimate new stressors with little labeled data. A fundamental breakthrough in our approach lies in the inclusion of the meta-knowledge inheritance mechanism, which equips our model with the ability to prevent catastrophic forgetting when adapting to new stressors. The experimental results show that our model achieves state-of-the-art performance compared with the baselines. Additionally, we construct a social media-based stressor estimation dataset that can help train web mining models to facilitate human well-being.

## CCS CONCEPTS

• **Human-centered computing** → *Social media*; • **Applied computing** → *Psychology*.

## KEYWORDS

stressor estimation, social media, meta-knowledge inheritance

**ACM Reference Format:**
Anonymous Author(s). 2018. MISE: Meta-knowledge Inheritance for Social Media-Based Stressor Estimation. In *Proceedings of Make sure to enter the correct conference title from your rights confirmation emai (Conference acronym 'XX)*. ACM, New York, NY, USA, 11 pages. https://doi.org/XXXXXXX.XXXXXXX

**Figure 1: Two posts. The first user's stress is caused by exams. The second user's stress is caused by writing paper. While prior studies may classify both posts as belonging to the broader category of school-related stress, our work aims to estimate the specific causes of stress (i.e., *exams* and *writing paper*), in order to provide more targeted support for stress relief.**

## 1 INTRODUCTION

With the rapid development of economy and society, people are under unprecedented psychological stress, coming from various aspects of life. As excessive stress without timely relief can negatively affect people's thoughts, feelings, behaviors, and physical and mental health [10], estimating and managing stress have become a big issue in the contemporary society.

Beyond traditional counseling and questionnaires based stress detection methods [3, 6, 12, 17], leveraging social media for stress detection has gained considerably increased attention in recent years. Through analyzing people's free-styled linguistic expressions and social behaviors, it is feasible to automatically and timely detect stress.

So far, the majority of social media-based stress detection work focuses on classifying user's **stress state** (i.e., stressed or non-stressed) [13, 14, 48, 50] and **stress category** (i.e., stress is classified into several broader categories, such as school, work, financial state, etc.) [22, 25, 56].

The aim of this study is to go further and identify users' specific **stressors**. We argue that in order to provide effective treatment for stress relief, it is necessary to have a thorough understanding of the specific causes of stress. Psychological research has demonstrated that stress arises from stressors [27]. Figure 1 shows two posts, from which we can know that the first user's stress comes from *exams*, and the second user's stress comes from *writing paper*. Accurately identifying stressors such as *exams* and *writing paper* is crucial in offering targeted support for stress relief.

Nevertheless, building an effective stressor estimation framework is non-trivial, facing two typical challenges. 1) First, users' stressors are quite diverse. Data for model's training exhibits the characteristic of having many different classes but a few examples

per class. 2) Second, new stressors incessantly appear as time progresses (e.g., covid-19 is a stressor appearing after 2019). The model must have the ability to learn the latest stressors quickly.

As traditional supervised learning methods require a large amount of data for supervised training to recognize each stressor, they are not suitable for new stressors and long-tailed stressors with scarce data. Hereby, we cast the problem of social media-based stressor estimation within a few-shot learning setting. Few-shot learning intends to train a deep learning model to recognize new classes with only a few labeled training examples, given prior experience with very similar tasks for which we have large training examples available [45, 47]. Previous work has suggested an effective way to acquire knowledge from a few examples via meta-learning [18, 38, 53, 58]. Meta-learning (often described as "learning to learn") advocates to learn at two levels, each associated with different time scales. The first is to quickly acquire task-specific knowledge within each separate task, and the second is to slowly summarize different task-specific knowledge to get the generic knowledge across the tasks. With the internal representation that is broadly suitable for many tasks, the obtained meta-model can thus quickly adapt to new environments through fine-tuning with a few labeled samples when facing an unseen task.

This motivates us to build a meta-learning based stressor estimation framework that is trained with past data, and can effectively be adopted to estimate new stressors with only a small number of training examples. Inspired by human's fast learning ability of entering the task with a large amount of prior knowledge encoded in the brains and DNA [37], we enhance meta-learning with a **meta-knowledge inheritance mechanism**, which defines a novel meta-knowledge inheritance loss and revised overall training objective to inherit knowledge from the prior meta-model without catastrophic forgetting for better model adaption.

In summary, the paper makes the following three contributions.

- From the **task** perspective, we propose a new task aimed at enhancing human well-being: social media-based *practical scenario few-shot stressor estimation* task. Unlike previous stress classification tasks, this task focuses on estimating specific causes of stress, enabling us to provide more targeted support for stress relief. We define this task with challenges that are practical.
- From the **method** perspective, we introduce a novel meta-learning-based stressor estimation framework, which incorporates a specially designed *meta-knowledge inheritance mechanism*. Our model exhibits strong generalization ability, enabling it to estimate new stressors with a little labeled data and avoid the problem of catastrophic forgetting.
- From the **data** perspective, we create a *stressor-oriented dataset* that contains 4,254 manually annotated posts. Our publicly available data would enable future research facilitating human well-being in different fields. The dataset and code will be released on publication [1].

The performance study on the constructed dataset shows that the proposed stressor estimation framework can achieve over 74.2% F1-score, which significantly outperforms both traditional and few-shot sequence labeling baselines.

---

[1]anonymous (Note that applicants need to sign an agreement about ethics)

As stress-causing health problems have continued to increase all over the world, we hope this work could stimulate further interests in leveraging social media as data sources and web mining approaches to help address this critical issue.

## 2 RELATED WORK

### 2.1 Stress Detection on Social Media

**Stress State Classification** These studies aim to classify whether an individual is stressed or not and the level of stress [26, 39]. Xue et al. [61] proposed a framework for chronic stress detection by aggregating individual tweet's stress detection results. Saha and Choudhury [39] presented machine learning techniques to assess how the stress of campus population changes following an incident of gun violence. TensiStrength [48] employed a lexical approach and a set of rules to classify direct and indirect expressions of stress or relaxation. Based on this, Gopalakrishna et al. [13] introduced word sense disambiguation by word sense vectors to improve the performance of TensiStrength. Guntuku et al. [14] explored multiple domain adaptation algorithms to adapt user-level Facebook models to Twitter language. Wang et al. [57] studied personalized stress classification step by step from the generic mass level, group level, to final individual level. Turcan et al. [50] explored multi-task learning to co-train stress classification with emotion classification. Alghamdi et al. [1] instructed the large language model to classify stress state based on post content summary.

**Stress Category Classification** These studies aim to classify individual's stressful states in a preset stress category, e.g., study, work, and financial state [65]. Xue et al. [60] investigated a number of features and employed Gaussian Process to classify stress in different categories. Based on this, Lin et al. [23] further introduced image feature and social interaction feature to improve performance. Zhao et al. [65] considered content, posting, interaction, and comment-response features to detect stress category with support vector machine. Lin et al. [25] proposed a multi-task convolutional neural networks model to classify stress category and subject. Li et al. [22] built five stress-related lexicons corresponding to the five stress categories and employed a Chinese natural language processing tool to analyze stress. Cao et al. [4] pre-trained BERT with a stress post classification task and proposed a multi-attention model to detect chronic stress in each category. Wang et al. [56] proposed to classify rarely appeared stress categories with GCN and Mixture of Experts mechanism.

Although these studies have explored states and categories of stress, they have not estimated specific causes of stress, which are critical to subsequent stress relief. Therefore, we further propose to leverage social media for stressor estimation. Moreover, prior studies are scarcely feasible in real world with incessantly emerging new stressors. They rely on a predefined set of categories, being unable to adapt to latest stressors. In this paper, we propose a meta-learning stressor estimation framework. Specifically, The meta-learning process allows the framework to be trained on the past time periods and quickly applied to the latest time period with a few labeled samples.

## 2.2 Sequence Labeling

Sequence labeling (SL) [41] is the process of efficiently assigning labels to each individual words within a given sequence. A common labeling formats used in this context is BIOES [62], where 'B' signifies 'Begin', 'I' denotes 'Intermediate', 'O' represents 'Other', 'E' corresponds to 'End', and 'S' indicates 'Single'. In the early stages of research, this task heavily relied on manually engineered features and conditional random fields [20]. However, recent advancements in deep learning have ushered in a revolution in sequence labeling. Modern models like Bidirectional LSTMs [19], Transformers [52], and BERT-based architectures [51] have consistently achieved state-of-the-art performance, transforming the landscape of sequence labeling and extending its applicability to various natural language processing tasks.

Named Entity Recognition (NER) [34] is a common sequence labeling task, aimed at identifying and categorizing entities, such as names of persons, organizations, and locations, within text [33, 35, 42, 43]. However, NER tasks typically involve the identification and further classification of named entities, which differs from the objectives of our application-oriented stressor estimation. Our focus lies in pinpointing the specific causes of stress to enable targeted stress relief, obviating the need for further classification.

Our work focuses on the real-world stressor estimation task, which presents unique challenges beyond traditional SL or NER tasks. 1) In the context of social media, stressors are more diverse, casual, and personalized, encompassing not only named entities but also gerunds (e.g., partying) and phrases (e.g., working overtime). The lack of strict structure and diverse linguistic expressions in social media posts necessitates the exploration of robust and adaptable natural language processing techniques. 2) Furthermore, our primary concern is to address the issue of catastrophic forgetting that arises when new stressors are learned over time. Handling this continual learning scenario is non-trivial and requires careful consideration in designing our method. This stands in contrast to most few-shot SL or NER tasks, which typically concentrate solely on performance improvements on new classes.

## 2.3 Meta-learning

In the literature, meta-learning has demonstrated good strength in dealing with few-shot learning problems. It learns the common parts of different tasks, and then adapts the obtained meta-model to new tasks rapidly with a few training examples [58]. Meta-learning approaches generally fall into three categories: model-based, metric-based, and optimization-based.

(1) Model-based meta-learning intends to design a meta-learner model that can update the parameters rapidly with a few training steps [36, 40]. For instance, Santoro et al. [40] trained a Memory-Augmented Neural Network as a meta-learner, which accumulated knowledge about previous tasks in an external memory that can fastly adapt to new tasks. (2) Metric-based meta-learning learn a generalized metric to measure the distance between samples in a feature embedding space [45, 47, 54]. For instance, Prototypical Network [45] averaged the embeddings of the labeled examples as the prototype for every class, and then measured the distance

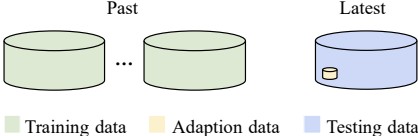

Figure 2: Schematic diagram of our practical scenario few-shot stressor estimation task, which focuses on the practical scenario that new stressors continue to emerge over time and some long-tailed stressor has scarce labeled data. Specifically, this scenario needs to train models with past data and effectively estimate latest stressor with a few adaption data.

between each test instance and each prototype. (3) Optimization-based meta-learning strives to learn well-initialized model parameters that can generalize better to similar tasks [8, 9, 37, 38]. For instance, MAML [8] trained model's parameters through double gradient updates, so that several gradient steps with a few labeled samples can achieve good results on new tasks. Among these three approaches, the optimization-based approach is more suitable for our problem. This is because the optimization-based approach is model structure-independent. Unlike the model-based and distance-based approaches, it does not require an additional memory module to store historical knowledge or a specialized relational module to compare the query set with the support set.

In this study, we build a meta-learning-based stressor estimation framework. To avoid catastrophic forgetting [63] when learning new stressors, we expand upon the optimization-based meta-learning by introducing a meta-knowledge inheritance mechanism. Our approach exhibits significant differences from prior meta-learning studies, as we define a novel meta-knowledge inheritance loss and a revised overall training objective for better meta-model adaption.

## 3 PROBLEM DEFINITION

Given a post $X = \{x_1, x_2, \cdots, x_n\}$, where $x_i$ denotes the $i$-th word in the post. Targeting to identify specific stressors that cause user stress, stressor estimation can be generalized as a sequence labeling problem. To achieve this, our framework needs to learn a mapping function $F(\phi) : X \rightarrow Y$ that takes $X = \{x_1, x_2, \cdots, x_n\}$ as the input and output the corresponding label sequence $Y = \{y_1, y_2, \cdots, y_n\}$ encoded with BIOES (B-Begin, I-Intermediate, O-Other, E-End, S-Single). For instance, given the input sequence "The dual stress of losing job and buying a house make me almost vomit blood!", we can obtain stressors "losing job" and "buying a house" from the corresponding label sequence "O,O,O,O,B,E,O,B,I,E,O,O,O,O,O,O".

In this paper, we focus on the practical scenario that new stressors continue to emerge over time and some long-tailed stressor has scarce labeled data. This requires the capability of learning to identify such stressors with just a few labeled samples. We represent the posts from past time periods with $D_p = \{d_1, d_2, \cdots, d_m\}$ and the posts from the latest time period with $D_l = \{d_{m+1}\}$. Here, $d_i = \{X_1, X_2, \cdots, X_a\}$ represents the post set within the $i$-th time period, and we define a half-year as a time period[2].

---

[2]The selection of the half-year time period was based on its unique ability to strike a balance between capturing both historical stressors and new stressors. By choosing a

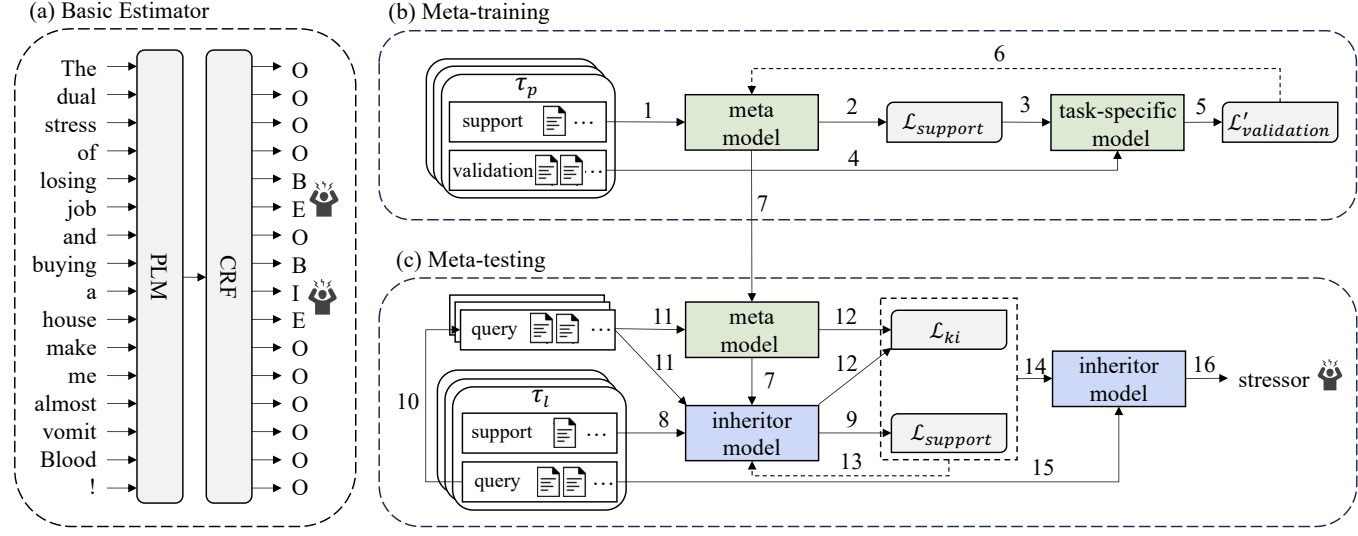

**Figure 3: Overview of our proposed framework. (a) The basic estimator. (b) The meta-training process. (c) The meta-testing process with meta-knowledge inheritance mechanism.**

The aim of our work is to build a stressor estimation framework, which is trained on past time periods $D_p$ and able to quickly adapt to estimate new stressors on latest time period $D_l$. However, due to the limited labeled support set in $D_l$, optimizing a satisfactory supervised learning framework is challenging. To address this, we perform meta-learning on $D_p$ to estimate generic knowledge that can be applied to perform better few-shot learning. To facilitate this process, we design a meta-knowledge Inheritance mechanism that inherit prior knowledge to prevent catastrophic forgetting.

## 4 METHODOLOGY

Figure 3 illustrates the overall framework of our meta-learning based stressor estimation enhanced with meta-knowledge inheritance mechanism. Our framework contains three parts: basic estimator, meta-training process, and meta-testing process. The basic estimator is employed to estimate stressors on social media. The meta-training process is designed to get a meta-model which learns the generic stressor contexts. The meta-testing process is designed to test our model for identifying new and unseen stressors with a few labeled samples. In this process, we propose a meta-knowledge inheritance mechanism to avoid catastrophic forgetting through inheriting knowledge from the prior meta-model.

### 4.1 Basic Estimator

The basic estimator (Figure 3.(a)) is responsible for post encoding and stressor estimation. Since pre-trained language models (PLMs) have strong semantic and context learning capabilities, we employ an effective pre-trained language model RoBERTa [28] to encode our post. Specifically, we input post $X = \{x_1, x_2, \cdots, x_n\}$ into RoBERTa to obtain the text representation $H = \{h_1, h_2, \cdots, h_n\}$.

---

half-year time period, we ensure that each interval contains a significant amount of historical stressor data, allowing us to analyze the ability of learning common stressor. Simultaneously, it also incorporats recent stressors, which are vital in assessing the system's ability to adapt and learn from new stressors.

Constraints are needed to ensure that the predicted label sequences are reasonable, such as ensuring that there will be no prediction of "...B,B..." (i.e., two begin labels are connected in the prediction label sequence). Since Conditional Random Field (CRF) [20] is effective to learn these constraints of the label sequences, we employ CRF to decode:

$$p(Y \mid H) = \frac{\exp\left(\sum_{i=0}^{\alpha} U(y_i, h_i) + \sum_{i=0}^{\alpha-1} T(y_i, y_{i+1})\right)}{\sum_{y' \in \mathcal{Y}(H)} \exp\left(\sum_{i=0}^{\alpha} U(y_i', h_i) + \sum_{i=0}^{\alpha-1} T(y_i', y_{i+1}')\right)}, \quad (1)$$

where $U$ denotes the emission function that represents the probability of predicting $y_i$. $T$ is the transition matrix which represents the probability that a transition from $y_i$ to $y_{i+1}$ occurs. $\mathcal{Y}(H)$ denotes the set of all possible label sequences for $H$. $Y$ is the corresponding label sequence to post $X$. The loss function is defined as:

$$\mathcal{L} = -\log p(Y \mid H), \quad (2)$$

### 4.2 Meta-learning Enhanced with Meta-knowledge Inheritance Mechanism

Since new stressors emerge as time progress and their labeled data are always limited, our model is expected to learn to estimate new stressors with a few labeled samples. Therefore, a meta-learning process (Figure 3.(b) and (c)) with our specially designed meta-knowledge inheritance mechanism is deployed.

*4.2.1 Meta-Training.* The meta-training is deployed on past time periods $D_p$ with relatively large number of labeled data. For each training epoch, a meta-task $\tau_p$ is constructed by randomly sampling a time period $d_i$ from the training set $D_p$, and the $K$ labeled samples in $d_i$ are chosen to serve as the support set $S$, as well as a part of remainder of $d_i$ are chosen to act as the validation set $V$.

Inspired by MAML [8], the two-level learning are applied to train a meta-model that can be quickly adapted to unseen new tasks. Let

$\theta$ denote the model parameter. For each meta task $\tau_p$, we first feed support set $S$ to the model and calculate the loss $\mathcal{L}_\tau(\theta)$ on $S$ to update $\theta$ through gradient descent:

$$\theta_\tau' = \theta - \alpha\nabla_\theta\mathcal{L}_\tau(\theta), \tag{3}$$

where $\alpha$ is the learning rate of first level meta-learning. $\theta_\tau'$ denotes the updated model parameter. This first process is to update the task-specific model, which quickly acquires task-specific knowledge (such as remembering the specific stressors and their context) through minimizing the loss on $S$. Secondly, we update the meta-model through the loss $\mathcal{L}_\tau(\theta_\tau')$ on validation set $V$,

$$\theta \leftarrow \theta - \beta\nabla\theta \sum_{\tau\in\mathcal{T}} \mathcal{L}_\tau(\theta_\tau'), \tag{4}$$

where $\beta$ is the learning rate of the second level meta-learning. This second process is to get the meta-model, which slowly summarizes task-specific knowledge as generic knowledge (such as summarizing generic stressor context) through minimizing the loss on $V$. After sufficient training over meta-training tasks, the meta-model can be quickly adapted for stressor estimation over meta-testing tasks.

*4.2.2 Meta-Testing.* The meta-testing is performed on the latest time period $D_l$ with a few labeled samples. we apply the same epoch-constructed mechanism to test whether our model can indeed adapt quickly to estimate the latest stressor. For each testing epoch, we construct a meta-task $\tau_l$ by randomly sampling the support set $S$ and query set $Q$ from $D_l$. The support set is employed to optimize the meta-model for adapting the new task. The query set is to test the optimized meta-model. The result is defined as the average performance across all testing epochs.

**Meta-knowledge Inheritance Mechanism.** When adapting the meta-model to a new testing task with a few labeled samples, it will suffer from the problem of **catastrophic forgetting** [11], i.e., the learning of a new task may cause the model to forget the knowledge learned from previous tasks. We argue that prior knowledge is important for estimating stressors in the latest time periods. Because in addition to new stressors and rare stressors, there are also some common stressors (e.g., exam) in the latest time period. Knowledge from previous tasks can help our model effectively estimate these common stressors, so it is essential to avoid catastrophic forgetting.

We propose a meta-knowledge inheritance mechanism to generate a inheritor-model, which consolidates the knowledge from the meta-model when adapting to new testing tasks. Specifically, we define a knowledge inheritance loss for meta-knowledge consolidating. The core idea of the knowledge inheritance loss is to make the inheritor-model imitate the prediction results of the meta-model on the query set $Q$. These prediction results are represented by soft labels [16], which are formulated as the predicted stressor label probability:

$$P_b^M(e_i, t) = \frac{\exp[f_b(e_i/t)]}{\sum_{c=0}^{4}\exp[f_c(e_i/t)]}, \tag{5}$$

where $e_i$ denotes the representation of word $x_i$ in post $X_j$ from query set $Q$ for meta-testing task $\tau_l$. $t$ is the temperature parameter to soften the peaky softmax distribution. $f_c(e_i)$ represents the logit

**Algorithm 1** Meta-learning Enhanced with meta-knowledge Inheritance Mechanism

---
**Require:** $\alpha, \beta$: the learning rates.
  **while** not done the meta-training **do**
    Sample batch of meta-task $\tau_p$ from $D_p$
    **for** Each $\tau_p$ **do**
      Sample a support set $S$ and a validation set $V$
      Evaluate $\mathcal{L}_\tau(\theta)$ with data $S$
      Compute task-specific parameters with gradient descent: $\theta_\tau' = \theta - \alpha\nabla_\theta\mathcal{L}_\tau(\theta)$
    **end for**
    Updating meta-model's parameters with data $V$ from each meta-task: $\theta \leftarrow \theta - \beta\nabla\theta \sum_{\tau\in\mathcal{T}}\mathcal{L}_\tau(\theta_\tau')$
  **end while**
  **while** not done the meta-testing **do**
    Sample batch of meta-task $\tau_l$ from $D_l$
    **for** Each $\tau_l$ **do**
      Sample the support set $S$ and the query set $Q$
      Evaluate $\mathcal{L}(\theta)$ with data $S$
      Inherit knowledge from the meta-model with meta-knowledge Inheritance loss:
      $P_b^M(e_i, t) = \frac{\exp[f_b(e_i/t)]}{\sum_{c=0}^{4}\exp[f_c(e_i/t)]}$,
      $\mathcal{L}_{ki}(\theta) = t^2 \sum_{X_j}\sum_{e_i}\sum_{c=0}^{4}P_c^M(e_i, t)\log\left(\frac{P_c^M(e_i,t)}{P_c^T(e_i,t)}\right)$
      Calculate the overall loss: $\mathcal{L}_{\text{total}}(\theta) = (1-\lambda)\mathcal{L}(\theta) + \lambda\mathcal{L}_{ki}(\theta)$
      Update the inheritor-model's parameters via optimizing Equation: $\theta^* = \theta - \alpha\nabla_\theta\mathcal{L}_{\text{total}}(\theta)$
    **end for**
  **end while**

---

score that $e_i$ achieves on label $c$. The knowledge inheritance loss is defined as follows:

$$\mathcal{L}_{ki}(\theta) = t^2 \sum_{X_j}\sum_{e_i}\sum_{c=0}^{4}P_c^M(e_i, t)\log\left(\frac{P_c^M(e_i, t)}{P_c^T(e_i, t)}\right), \tag{6}$$

where $P_c^M$ and $P_c^T$ denote the predicted distributions of meta-model and inheritor-model, respectively. At the same time, we employ the CRF loss on support set $S$ to make inheritor-model learn task-specific knowledge. Then, we provide the overall learning objective $\mathcal{L}_{\text{total}}(\theta)$, which is a summation of knowledge inheritance loss $\mathcal{L}_{ki}(\theta)$ on query set and the CRF loss on support set as follows:

$$\mathcal{L}_{\text{total}}(\theta) = (1-\lambda)\mathcal{L}(\theta) + \lambda\mathcal{L}_{ki}(\theta), \tag{7}$$

where $\lambda$ denotes a trade-off parameter for the two losses. The inheritor-model for meta-testing tasks is initialized with the meta-model obtained by meta-training, and further updated as follows:

$$\theta^* = \theta - \alpha\nabla_\theta\mathcal{L}_{\text{total}}(\theta), \tag{8}$$

where $\alpha$ denotes the learning rate. Finally, the inheritor-model is deployed to estimate stressors in the query set of meta-testing tasks. We summarize the meta-training and meta-testing process as shown in Algorithm 1.

**Summary.** Our approach exhibits significant differences from existing meta-learning studies, as we define a novel meta-knowledge inherit loss and a revised overall training objective for better meta-model adaption. In other words, equations (5)(6)(7)(8) in our approach are core innovations and differences compared with existing meta-learning approaches.

# 5 EXPERIMENT

## 5.1 Dataset Construction and Statistics

One significant challenge for this new task is the lack of available dataset. The most relevant datasets we discovered, such as Dreaddit [49] for stress state detection and Plscd [56] for stress category classification, only provide labels indicating whether a post expresses stress and its broad category (e.g., work, study, financial state). Unfortunately, these datasets do not include the specific stressor labels that are necessary for our stressor estimation task. To overcome this challenge, we build a social media-based stressor estimation dataset. In order to facilitate web mining research for human well-being, we will release our dataset on publication[3].

The dataset is crawled from Weibo. Since Weibo is one of the most popular social media platforms worldwide, with 249 million daily active users [44]. Specifically, the data labeling process is carried out as follows:

Firstly, we employ the retrieval function to get the stressful posts. The retrieval term is user's self-report, such as "I, stressed". This self-report ensures that the posts contain stress expression [24].

Then, we hire six master students who major in psychology and are active on social media to annotate the stressor. Annotators are asked to read posts carefully and understand the semantics of posts to label stressors. We pay annotators $0.15 per post. There are situations in which the user only expresses that he/she is stressed without mentioning the source of the stress. In this case, no annotation is made. Note that the annotators are asked to further remove the following types of noise data:

- Others feel stressed, e.g., "*Is it really stressful to chat with me? Why are they so easily nervous?*"
- Advertise, e.g., "*#Loan# #Internet Loan# #Universal Gold# Can't pay your monthly bills? Repayment stress? Poke me to get the universal gold*"
- Stress of the past, e.g., "*Every time I pass by here, I think of the time when I was preparing for the postgraduate entrance examination. At that time, I was so stressed.*"
- Future plan, e.g., "*I hope that in the future I will never get married because of age and stress!*"
- Lyrics or verses, e.g., "*Some days, I just wanna leave the negativity in my head. I just want relief from my stress (Song Lyrics)*"

Each post is analyzed by three different annotators. Only if their annotated stressors are consistent, the stressor is used as the annotation result. If their annotated stressors are inconsistent, all different stressors are kept pending further annotation.

Finally, the annotation progress is completed by a psychology expert from the Department of Psychology. The main work of the expert can be summarized in two parts: 1) checking the posts with consistent annotation results and 2) giving an expert opinion as the final annotation for those posts with inconsistent annotation.

In total, we obtain 4,254 labeled posts from June 2018 to June 2022. The average inner-agreement (i.e., Cohen's Kappa) between annotators is 0.71, indicating good agreement. The data set is divided into different time periods based on half-year intervals. We have seven past time periods, labeled as $D_p$, spanning from June

---

[3]Our dataset could be requested on https://github.com/anonymous

2018 to December 2021. Additionally, we have one latest time period, denoted as $D_l$, covering January 2022 to June 2022.

## 5.2 Baselines

We compare our method with the following traditional sequence labeling baselines:

- LSTM-CRF [19]. A classical sequence labeling method with bidirectional LSTM and Conditional Random Field.
- CNN-CRF [32]. Another classical sequence labeling with bidirectional LSTM, CNN, and CRF.
- SED [2]. It employs attention mechanism to get text representation and LSTM to decode the representation.
- DetIE [51]. It freezes part of BERT and trains the rest part combined with fully connected layer.
- CRUP [29]. It enhances the representations through contrastive learning for sequence labeling.

These traditional baselines are not specifically designed for few-shot learning. For fair comparison with our meta-learning model, all these baselines also only take $K$ samples (i.e., support set in meta-task) to learn to estimate stressors in the latest time period. Specifically, we utilize all the past time periods data $D_p$ in the training stage. Then the trained model is fine-tuned with $K$ samples from $D_l$. Lastly, we test the fine-tuned model on the latest time period $D_l$ (except the $K$ samples).

We consider a practical scenario that new stressors continue to emerge over time and some long-tailed stressor has scarce labeled data. Thus we employ the following few-shot sequence labeling baselines to further verify the effectiveness:

- SimBERT [7]. It employs BERT to get token representation and predict labels through finding the most similar labeled token in the support set.
- Ma21 [31]. It casts sequence labeling as machine reading comprehension problem and generates the labeling types as questions.
- ConVEx [15]. It proposes a pairwise cloze task to pre-training model with Reddit data and then fine-tune the pre-trained model with a few labeled samples.
- ESD [55]. It formulates the sequence labeling problem as classification of each span and combines the idea of Prototypical Network [45] for classification.
- BDCP [59]. It introduces entity boundary discriminative module to provide a highly distinguishing boundary representation space for labeling.

## 5.3 Experiment Details

For each meta-task of meta-learning, we sample 3, 5, or 10 posts as the support set (i.e., 3-shot, 5-shot, or 10-shot) and 15 posts as the validation or query set. The first level learning rate $\alpha$ is set to 2e-5. The second level learning rate $\beta$ is 5e-5. The max training epoch is set to 5000. We report the average performance from 50 random testing epochs. AdamW [30] is adopted as our optimizer. 10% drop out [46] is deployed to avoid overfitting. The temperature parameter $t$ is set to 5 and the trade-off parameter $\lambda$ is set to 0.2. Experiments are run on a Linux server with RTX 2080 GPUs. Pytorch 1.71 is used to construct the models. We employ transformers 4.18 to load RoBERTa [28] without pooling layer as PLM. Our model has

**Table 1: Main Results. The first part lists the performance of traditional sequence labeling baselines. The second part shows the performance of few-shot baselines. MISE denotes our method.**

| Method | 3-shot | | | 5-shot | | | 10-shot | | |
|---|---|---|---|---|---|---|---|---|---|
| | Precision | Recall | F1-score | Precision | Recall | F1-score | Precision | Recall | F1-score |
| LSTM-CRF [19] | 0.5684 | 0.5870 | 0.5775 | 0.5674 | 0.6081 | 0.5829 | 0.5778 | 0.6298 | 0.5986 |
| CNN-CRF [32] | 0.5643 | 0.6126 | 0.5862 | 0.5752 | 0.6134 | 0.5958 | 0.5895 | 0.6330 | 0.6014 |
| SED [2] | 0.5761 | 0.6310 | 0.6023 | 0.5891 | 0.6385 | 0.6056 | 0.5982 | 0.6507 | 0.6311 |
| DEtIE [51] | 0.5996 | 0.6333 | 0.6164 | 0.6055 | 0.6503 | 0.6238 | 0.6126 | 0.6854 | 0.6538 |
| CRUP [29] | 0.6007 | 0.6364 | 0.6196 | 0.6120 | 0.6528 | 0.6263 | 0.6230 | 0.6925 | 0.6563 |
| SimBERT [7] | 0.6086 | 0.6417 | 0.6248 | 0.6359 | 0.6791 | 0.6543 | 0.6538 | 0.6910 | 0.6766 |
| Ma21 [31] | 0.6274 | 0.6204 | 0.6239 | 0.6463 | 0.6310 | 0.6384 | 0.6742 | 0.6601 | 0.6671 |
| ConVEx [15] | 0.6216 | 0.6422 | 0.6317 | 0.6457 | 0.6673 | 0.6562 | 0.6731 | 0.6956 | 0.6881 |
| ESD [55] | 0.6413 | 0.6612 | 0.6509 | 0.6582 | 0.6897 | 0.6745 | 0.6814 | 0.7020 | 0.6902 |
| BDCP [59] | 0.6360 | 0.6573 | 0.6459 | 0.6498 | 0.6759 | 0.6627 | 0.6734 | 0.6948 | 0.6855 |
| **MISE (Ours)** | **0.6825** | **0.7029** | **0.6914** | **0.7053** | **0.7366** | **0.7212** | **0.7332** | **0.7541** | **0.7420** |

**Table 2: Ablation Study. M denotes the entire meta-learning process. I denotes the meta-knowledge inheritance mechanism.**

| Method | 3-shot | | | 5-shot | | | 10-shot | | |
|---|---|---|---|---|---|---|---|---|---|
| | Precision | Recall | F1-score | Precision | Recall | F1-score | Precision | Recall | F1-score |
| **MISE** | **0.6825** | **0.7029** | **0.6914** | **0.7053** | **0.7366** | **0.7212** | **0.7332** | **0.7541** | **0.7420** |
| w/o M | 0.6178 | 0.6403 | 0.6289 | 0.6419 | 0.6743 | 0.6570 | 0.6689 | 0.6843 | 0.6785 |
| w/o I | 0.6392 | 0.6713 | 0.6596 | 0.6628 | 0.6954 | 0.6807 | 0.6975 | 0.7182 | 0.7068 |

101.77M parameters. The total computational time is 2.2 hours. As we want to estimate the specific stressor with clear boundaries, we employ token-level precision, token-level recall, and token-level F1 score to measure the performance.

## 5.4 Main Results

Table 1 reports the performance of all methods. The first part shows the performance of traditional sequence labeling methods. The average performance of these methods is lower than 63%, which indicates that these methods are not good at estimating new stressor with a few labeled samples.

The second part lists the performance of few-shot baselines. We can see that they outperform traditional methods with 5% average improvement, which suggests that the meta-learning or few-shot learning process can significantly enhance the estimation performance in the latest time period.

Our method MISE achieves over 74.2% in F1-score, with over 4.0-5.1% improvement compared with the best baseline. It illustrates that our meta-learning method is effective in estimating new stressor with a few labeled samples. We attribute the improvements to the fact that our method is well-designed with the meta-learning process and meta-knowledge inheritance mechanism. Furthermore, our model's standard deviation is 0.012, 0.013, and 0.019 for 3-shot, 5-shot, and 10-shot settings, indicating good robustness.

There is a general trend of improvement in performance as we move from traditional baselines to few-shot baselines, and then to our MISE method. The improvement in F1-scores from 3-shot to

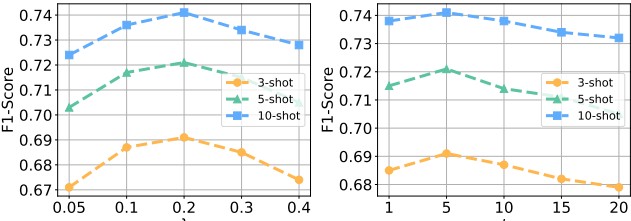

**Figure 4: Parameter Study on $\lambda$ and $t$.**

10-shot learning scenarios suggests that having more support set examples improves the model's performance, which is a common trend in few-shot problem.

## 5.5 Ablation Study

To analyze the effect of different components in MISE, we construct ablation experiments as shown in Table 2. The performance of MISE will drop over 6.2% without the entire meta-learning process, which verifies that meta-learning can effectively make our model learn to identify new stressors with a few labeled samples. The average decline in performance will be 3.5% without the meta-knowledge inheritance mechanism. This demonstrates that meta-knowledge inheritance enables our model to effectively avoid catastrophic forgetting and enhance model performance.

**Table 3: Catastrophic Forgetting Study. All methods are tested on the past time periods after adaptation to latest time period.**

| Method | Precision | Recall | F1-score |
|---|---|---|---|
| LSTM-CRF [19] | 0.6674 | 0.6643 | 0.6665 |
| CNN-CRF [32] | 0.6740 | 0.6628 | 0.6687 |
| SED [2] | 0.6921 | 0.6747 | 0.6854 |
| DelIE [51] | 0.7137 | 0.7141 | 0.7139 |
| CRUP [29] | 0.7110 | 0.7163 | 0.7135 |
| SimBERT [7] | 0.6942 | 0.6748 | 0.6836 |
| Ma21 [31] | 0.7039 | 0.6902 | 0.6965 |
| ConVEx [15] | 0.7217 | 0.7089 | 0.7141 |
| ESD [55] | 0.7175 | 0.7055 | 0.7106 |
| BDCP [59] | 0.7146 | 0.7074 | 0.7113 |
| **MISE (ours)** | **0.7952** | **0.8015** | **0.7983** |

**Table 4: Past Time Periods Performance Study. All methods are tested under traditional supervised scenario.**

| Method | Precision | Recall | F1-score |
|---|---|---|---|
| LSTM-CRF [19] | 0.7652 | 0.7578 | 0.7615 |
| CNN-CRF [32] | 0.7719 | 0.7616 | 0.7667 |
| SED [2] | 0.7835 | 0.7573 | 0.7702 |
| DelIE [51] | 0.8016 | 0.8034 | 0.8023 |
| CRUP [29] | 0.8028 | 0.8069 | 0.8054 |
| SimBERT [7] | 0.7904 | 0.7752 | 0.7781 |
| Ma21 [31] | 0.7761 | 0.7894 | 0.7827 |
| ConVEx [15] | 0.7993 | 0.8018 | 0.8006 |
| ESD [55] | 0.8025 | 0.8065 | 0.8047 |
| BDCP [59] | 0.8021 | 0.8053 | 0.8040 |
| **MISE (ours)** | **0.8182** | **0.8235** | **0.8206** |

## 5.6 Catastrophic Forgetting Study

To evaluate the effectiveness of MISE in addressing the catastrophic forgetting problem, we conduct an experiment that tests model's performance on previous tasks after it learns a new task. Firstly, we randomly select 20% of the data from past time periods $D_p$ and utilize 80% of the remaining data to train the model. Next, we adapt the model using a few labeled samples from latest time period $D_l$. Finally, we test the adapted model using the 20% isolated data. This process is repeated five times to calculate the average performance. As shown in Table 3, MISE outperforms the best baseline over 8.4% in F1-score. It verifies that our proposed meta-knowledge inheritance mechanism can effectively resolve the catastrophic forgetting problem when learning a new task.

## 5.7 Past Time Periods Performance Study

To verify MISE's effectiveness in estimating stressors from past time periods (supervised scenario), we design an experiment that trains and tests the models on the past time periods' data. Specifically, we conduct five-fold cross-validation on data from past time periods $D_p$. The results are shown in Table 4. MISE achieves over 81.8% performance, with over 1.5% improvement compared with the best

**Table 5: Case Study. There are two real-world posts. MISE successfully estimate a new stressor and a forgettable stressor.**

| |
|---|
| **Post**: I feel stressed most of the time. Monkeypox has made life not go well for anyone lately.

**MISE output**: Monkeypox |
| **Post**: I want to buy an iPad Pro... but I have to save money for the decoration of my new home. I am very stressed.

**MISE output**: decoration |

baseline. It illustrates that our framework can not only work better on the estimating latest stressors with a few labeled data but also be effective on estimating stressors from past time periods.

## 5.8 Parameter Study

To analyze the sensitivity of MISE, we report the performance with a variety of hyperparameters. Specifically, we analyze parameters $\lambda$ and $t$, corresponding to the trade-off for the overall loss and the re-scaling temperature for the meta-knowledge inheritance mechanism, respectively. As shown in Figure 4, $\lambda = 0.2$ and $t = 5$ are the best parameter settings. In addition, the performance of MISE keeps good with different $\lambda$ and $t$, which illustrates that MISE is relatively insensitive and robust to the changes of the settings.

## 5.9 Case Study

We provide insights based on real-world cases, as demonstrated in Table 5. In the first example, our framework correctly identified the user's stressor as "*monkeypox*". It is worth noting that "*monkeypox*" is a new stressor for the framework. This demonstrates that our meta-learning framework is capable of estimating the latest stressors with few samples. In the second example, our framework correctly identified the stressor as "*decoration*", which is an infrequently appeared stressor in the training data. In contrast, few-shot baselines fail to estimate the stressor "*decoration*", because these models forget the stressor "*decoration*" when adapting to the latest time period with few samples. This highlights the importance of our proposed meta-knowledge inheritance mechanism.

## 6 CONCLUSION

In this paper, we discover and propose a new social media-based stress-related task, i.e., stressor estimation. To address the practical problem of estimating the latest stressor with limited data, we propose a meta-learning framework that is trained on past data and can be adapted to the latest stressor estimation. We expand upon optimization-based meta-learning by introducing a meta-knowledge inheritance mechanism to avoid catastrophic forgetting when learning new stressors. Our approach defines a novel meta-knowledge inheritance loss and a revised training objective, which distinguishes it from prior meta-learning studies. Experimental results demonstrate that our framework is significantly effective when compared to all state-of-the-art baselines. In addition, a well-labeled social media stressor estimation dataset is proposed.

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

## A    LIMITATION

While much of the existing social media-related research assumes that users' expressions on social media are proactive and authentic [64], it is important to acknowledge that there are situations where users' posts may not accurately represent their innermost thoughts. For instance, a user's real stressor is "*interview*", but he/she might post: "*This bad weather makes me stressed.*" and not mention about "*interview*". In such cases, the effectiveness of our framework may be compromised.

## B    ETHICAL CONSIDERATIONS

Since this work is an intersection study of human psychology and data mining, the ethical issues must be carefully deliberated.

Privacy. All the data used in this paper is publicly available on social media. We acknowledge that the psychological experiments may potentially impact subjects [5], but our work is just an observation on public posts and there are no subjects.

Data protection. The dataset will be anonymized before being shared. We only provide post text without any personal information.

Applicants must sign an agreement before they get the dataset. This agreement guarantees: 1) they will never attempt to identify or contact any user in the dataset; 2) they will never make or use the dataset for commercial purposes; etc.

## C    APPLICATIONS

Our work has the potential to make a significant impact on society by improving the well-being of individuals.

1. Our work has potential applications in the field of psychological diagnosis. For instance, if a teenager experiences prolonged stress, they may be at an increased risk of self-harm or suicide. In such cases, our work can identify their stressor and alert their parents, enabling them to provide target care and support to prevent potential harm.

2. Our work provides new ideas and support for psychological research. Traditional psychology experiments and conclusions are often based on tens of hired subjects [21], with very low population sampling rates. Our social media-based approach can help them study psychology on a larger population scale. As a foundation, it can advance the field of psychology.

