# OpenReview forum: "MISE: Meta-knowledge Inheritance for Social Media-Based Stressor Estimation"
_ACM.org/TheWebConf/2025/Conference — WWW 2025 Oral_

### Official Review · Reviewer_pjkW · 2024-11-28

**Novelty:** 6
**Technical Quality:** 4

**Review:**

This paper proposes a fine-grained stressor estimation task, constructs a new dataset, and presents a novel stressor estimation approach that differs from existing methods.

Advantages:

(1) The paper proposes a novel social media-based stressor estimation task.

(2) The dataset construction process follows a rigorous validation logic with multi-round annotation.

(3) The paper proposes a meta-knowledge inheritance mechanism to address the catastrophic forgetting problem.

(4) The experimental settings are comprehensive with promising results.

Disadvantages:

(1) The experimental analysis is not comprehensive enough. For instance, there lacks experimental or theoretical justification for setting the time period to half-year;

(2) Some experimental details need to be more specific. For example, in Section 5.6, it is unclear whether the training process follows meta-learning or supervised learning. If it follows meta-learning, the performance being higher than Table 1 appears unreasonable;

(3) Although the dataset validation logic is rigorous, the paper lacks statistical analysis of stressor distribution in the dataset, which may not fully support the first challenge mentioned (having many different classes but a few examples per class);

(4) The dataset collection only considers Weibo platform, which may lead to platform bias and requires further discussion;

(5) According to Table 4, under the traditional supervised scenario, the model's advantage appears to be limited with only 1.5% improvement in F1-score, which lacks further investigation.

**Questions:**

Refer to the "Disadvantages" section in the review.

**Reviewer Confidence:**

4: The reviewer is certain that the evaluation is correct and very familiar with the relevant literature

**Scope:**

4: The work is relevant to the Web and to the track, and is of broad interest to the community

---

### Official Review · Reviewer_aBzc · 2024-11-30

**Novelty:** 5
**Technical Quality:** 4

**Review:**

The manuscript introduced a new stress detection task aimed at estimating more specific stressors through users’ posts on social media. Authors casted the problem within a practical scenario few-shot learning setting, and proposed a novel meta-learning based stressor estimation framework that is enhanced by a meta-knowledge inheritance mechanism.

Pros.: 1) The proposed model can not only learn generic stressor context through meta-learning, but also has a good generalization ability to estimate new stressors with little labeled data. 2) A social media-based stressor estimation dataset was built, based on which extensive experiemnts have been conducted, demonstrating that the proposed method significantly outperforms SOTA baselines.
Cons.: 1) Challenges of this work need be elaborated in more details, so as to better highlight the technical novelty of the work. 2) Compared with existing meta-learning approaches, what's the difference and contribution of the proposed method? 3) For new stressors (i.g., zero-shot cases), what the performance of the proposed method? 4) Including a section on future research directions to highlight potential extensions or applications of the proposed framework.

**Questions:**

1) Challenges of this work need be elaborated in more details, so as to better highlight the technical novelty of the work. 2) Compared with existing meta-learning approaches, what's the difference and contribution of the proposed method? 3) For new stressors (i.g., zero-shot cases), what the performance of the proposed method? 4) Including a section on future research directions to highlight potential extensions or applications of the proposed framework.

**Reviewer Confidence:**

3: The reviewer is confident but not certain that the evaluation is correct

**Scope:**

3: The work is somewhat relevant to the Web and to the track, and is of narrow interest to a sub-community

---

### Official Review · Reviewer_2C3W · 2024-12-02

**Novelty:** 3
**Technical Quality:** 3

**Review:**

The paper addresses an increasingly significant issue in modern society: detecting stress through social media posts. It introduces an innovative framework that leverages meta-learning supported by a meta-knowledge inheritance mechanism to estimate specific stressors (e.g., exam-related stress, work stress, etc.) based on user-generated content on social media. The model not only learns general stress context through meta-learning but also demonstrates good generalization ability, enabling it to estimate new stress sources even with limited labeled data.

Quality: The paper is well-structured. From the problem introduction to the proposed solution and experimental evaluation, the content is clear and logically organized. The method is clearly defined, and the use of the meta-learning framework is well-explained, allowing the research process to be clearly presented. However, the section on the meta-knowledge inheritance mechanism could benefit from more detailed explanations.

Clarity: The overall clarity of the paper is high. The introduction succinctly outlines the research problem and generates a need for more refined stress source classification methods. The background and related work sections are well-organized, providing a solid theoretical foundation for the proposed framework. However, certain technical details (such as the description of the meta-knowledge inheritance mechanism) might be too complex for non-specialist readers. It is recommended to include more diagrams or examples in these sections to help readers better understand the complex mechanism.

Originality: The paper successfully shifts the research focus from traditional stress classification to identifying specific stressors, such as exam stress or thesis-related stress. Furthermore, the integration of the meta-knowledge inheritance mechanism into the framework to address challenges like catastrophic forgetting is innovative in this field. This method is particularly valuable when dealing with new categories, as it prevents forgetting previous knowledge, enhancing the model's robustness and adaptability.

Significance: The significance of this paper seems somewhat limited, as it simplifies the handling of stress states. The article focuses on identifying stressors but overlooks the crucial role emotions play in stress experience. In reality, stress and emotions are often intertwined, and emotional fluctuations (such as anger, anxiety, sadness, etc.) significantly influence how individuals respond to and perceive stressors. Therefore, relying solely on stress states to infer users' mental states is somewhat one-sided and may not fully capture their emotional experiences. As a result, its application in areas such as mental health monitoring and personalized therapy might be limited.

Pros:
- The paper focuses on identifying specific stressors rather than broad stress categories, which may lead to more personalized and effective interventions.
- It presents a robust machine learning framework that combines few-shot learning and meta-learning, contributing effectively to the methodology. This combination ensures the model's ability to handle sparse data while alleviating the problem of catastrophic forgetting when learning new stress categories.

Cons:
- Data Diversity and Representativeness Issues: The dataset used in the paper may have limitations, especially in terms of handling data from different platforms and cultural backgrounds, which could introduce biases. Social media content is highly diverse, so ensuring the dataset's broad representativeness is an issue that requires further exploration. More extensive real-world validation is needed to confirm the model's generalizability across different populations and environments.
- Neglect of Emotions: Although the paper introduces an innovative method for stressor identification, the model focuses primarily on recognizing stress states while overlooking the critical role of emotional states in the experience of stress. In real-world scenarios, stress is often accompanied by emotional fluctuations. Emotions (e.g., anxiety, anger) and stress are interwoven and influence how individuals perceive and respond to stressors. Therefore, relying solely on the identification of stressors may fail to comprehensively capture an individual's mental state.
- Insufficient Contextual Understanding: While the model is capable of identifying specific stressors, the complexity of social media content may hinder its ability to fully understand the contextual background of posts. For instance, certain users might express stress through sarcasm, humor, or reverse expressions, which could lead to misinterpretations of their stress state. As such, the model may benefit from further improvements in context analysis and understanding.

**Questions:**

Handling Noise in Data Preprocessing: You mention using social media data to train your model. How do you deal with noise in the text during preprocessing, like spelling mistakes, emojis, slang, or non-standard language? When it comes to feature engineering, did you use specific strategies to clean and process the data, or did you feed the raw text directly into the model?

Model Performance Across Different Platforms: The experiments in your paper mainly focus on a dataset that you created. How does your model perform across different social media platforms like Twitter, Reddit, and Facebook? Given that content can vary significantly between platforms, does the model still perform consistently? Also, how does it handle different types of content, like short posts versus longer ones? And in the case of longer posts that might have nested structures, can the model still make accurate predictions?

Understanding Context and Situation: When identifying specific stressors, does the model fully understand the context behind a user's posts? For example, some posts may express stress because of the context (like a casual chat between friends). Can the model recognize when something is said in a joking or less serious way and avoid mistakenly classifying it as a real stressor?

**Reviewer Confidence:**

3: The reviewer is confident but not certain that the evaluation is correct

**Scope:**

3: The work is somewhat relevant to the Web and to the track, and is of narrow interest to a sub-community

---

### Official Review · Reviewer_QALN · 2024-12-03

**Novelty:** 6
**Technical Quality:** 5

**Review:**

-important topic
-interesting dataset that would be valuable for further research
-extensive baselines (though problem could possibly be framed differently, see my comment about zero-shot below)

issues:
-some of the content that should be part of the 8-page manuscript (e.g., limitations) are moved into the appendices. for this reason, the discussion is missing.
-introduction and related work sections should be streamlined
-the authors mention using one annotator but they also mention calculating Cohen's kappa---this is calculated between two annotators. So, it's unclear how intercoder reliability was tested.
-k=0.71 indicates challenges for data quality. it might be that the authors are trying to "predict opinions". Would like to hear the authors' response to this.
-catastrophic forgetting is not explained and it's not clear why this is a particular problem for stress detection
-could this problem be tackled with zero-shot class detection using LLMs like GPT models?
-"MISE achieves over 81.8% performance" >> F1 score is 0.821
-the paper is perhaps trying to accomplish too much which makes it challenging to report the experiments, results, and their implications at adequate depth. some trimming and decrease of scope could help.

**Questions:**

-some of the content that should be part of the 8-page manuscript (e.g., limitations) are moved into the appendices. for this reason, the discussion is missing.
-introduction and related work sections should be streamlined
-the authors mention using one annotator but they also mention calculating Cohen's kappa---this is calculated between two annotators. So, it's unclear how intercoder reliability was tested.
-k=0.71 indicates challenges for data quality. it might be that the authors are trying to "predict opinions". Would like to hear the authors' response to this.
-catastrophic forgetting is not explained and it's not clear why this is a particular problem for stress detection
-could this problem be tackled with zero-shot class detection using LLMs like GPT models?
-"MISE achieves over 81.8% performance" >> F1 score is 0.821
-the paper is perhaps trying to accomplish too much which makes it challenging to report the experiments, results, and their implications at adequate depth. some trimming and decrease of scope could help.

**Reviewer Confidence:**

2: The reviewer is willing to defend the evaluation, but it is likely that the reviewer did not understand parts of the paper

**Scope:**

4: The work is relevant to the Web and to the track, and is of broad interest to the community